# Offline Imagery Checks for Remote Drone Usage

**Roxane J. Francis** [1,*], **Kate J. Brandis** [1] **and Justin A. McCann** [2]

1 Centre for Ecosystem Science, School of Biological Earth and Environmental Sciences, University of New South Wales, Sydney, NSW 2052, Australia
2 Bush Heritage Australia, P.O. Box 329, Flinders Lane, Melbourne, VIC 8009, Australia
* Correspondence: roxane.francis@unsw.edu.au

**Abstract:** Drones are increasingly used for a wide range of applications including mapping, monitoring, detection, tracking and videography. Drone software and flight mission programs are, however, still largely marketed for "urban" use such as property photography, roof inspections or 3D mapping. As a result, much of the flight mission software is reliant upon an internet connection and has built-in cloud-based services to allow for the mosaicking of imagery as a direct part of the image collection process. Another growing use for drones is in conservation, where drones are monitoring species and habitat change. Naturally, much of this work is undertaken in areas without internet connection. Working remotely increases field costs, and time in the field is often aligned with specific ecological seasons. As a result, pilots in these scenarios often have only one chance to collect appropriate data and an opportunity missed can mean failure to meet research aims and contract deliverables. We provide a simple but highly practical piece of code allowing drone pilots to quickly plot the geographical position of captured photographs and assess the likelihood of the successful production of an orthomosaic. Most importantly, this process can be performed in the field with no reliance on an internet connection, and as a result can highlight any missing sections of imagery that may need recollecting, before the opportunity is missed. Code is written in R, a familiar software to many ecologists, and provided on a GitHub repository for download. We recommend this data quality check be integrated into a pilot's standard image capture process for the dependable production of mosaics and general quality assurance of drone collected imagery.

**Keywords:** UAV; ecology; conservation; remote area





## 1. Introduction

The utilisation of drones in a vast range of industries is rapidly increasing, alongside constant developments in drone technology. Many of these uses remain marketed for the "urban" world, where drones are used for industries such as urban design, insurance, forensics, and real estate, together with a huge market for hobbyist drone pilots [1–5]. These predominant uses of drones mean much of the accompanying flight mission software is reliant upon an internet connection to facilitate data collection and processing [6,7]. However, digital disadvantage in rural and remote areas means internet connectivity is limited in many continents, including Australia, Africa and South America [8–10].

Another increasing use of drones is their application in ecological monitoring. These applications include the monitoring of vegetation, locating of animals, or counting of large species aggregations [11–13]. The use of drones in natural environments lends itself to unique difficulties and requirements for the drone user. In undertaking an ecological monitoring survey, there are often no clear pre-existing boundaries that a pilot may need to cover in a drone flight. Further, boundaries can change or move quickly when surveying animals, and the development of survey areas is therefore often estimated on the fly. The estimation of aerial coverage based on on-ground distance calculations is difficult and as a result, complete coverage of a survey area in drone imagery may not be obtained.

Currently, there is no simple, free and/or open source way of evaluating flight quality and determining the geographical position of collected photographs while in the field.

Missing or wayward images can be a big problem when monitoring requires the creation of an orthomosaic (a georeferenced image derived from the mosaicking of many individual images) [14]. Orthomosaics are often crucial inputs into ecological monitoring as they allow for high-resolution mapping of study sites [15]. The creation of a successful orthomosaic (a complete and clear orthomosaic without gaps) requires the collection of consistently spaced images, captured along transect lines at a very high overlap (upwards of 75%). The collection of this imagery can take many hours in the field and can be prone to error due to software failures, incorrect camera settings, poor lighting, battery changes, GPS failures and weather conditions such as wind [16,17]. In the case where sections of flight paths are missed, the orthomosaic will not correctly align neighbouring images, resulting in a highly blurred or patchy final mosaic. Even if images are obtained, if they are not of sufficient quality (e.g., highly blurred), processing software is unable to locate tie points in neighbouring images, also resulting in blurred or missing portions of mosaics. Some drone flight mission programs such as Drone Deploy [18] and Pix4D Capture [19] have automated the process of developing orthomosaics by directly uploading imagery to the cloud for instant processing of the imagery. This allows for reasonably quick viewing of the placement of images in geographic space, but requires an internet connection and processing of the final mosaic can take several hours or days, making it difficult to evaluate collection quality. Other open source software such as WEB Open Drone Map is available at a cost and works offline [20], but as it is mosaicking software it requires large amounts of random access memory (RAM), a limitation when working on field-based computers or laptops. Further, many processing software licences are locked to a single computer and so the user must wait to return to the office, or attempt remote connection to then realise some images were not captured. As such, when working in the field without internet, quick viewing of the geographic placement of collected drone imagery is currently not possible, making it difficult to predict the success of later orthomosaic production in a disconnected environment.

Importantly, for many ecological drone surveys there is a very limited window in which surveys can be conducted as they are focused around a breeding, climatic, migration event, or season. The temporal aspect to the data collection is therefore highly important and repeat surveys are not possible once the opportunity has passed [21–24]. Where repeat surveys are possible, failure to collect enough suitable images for a complete orthomosaic can result in more days spent in the field or the rescheduling of field trips, increasing field costs. Failure to deliver an orthomosaic can result in failure to provide accurate, or any, monitoring data, making in-field checks highly useful.

We aimed to develop a totally offline, field friendly system of plotting drone image location in geographical space and quantifying potential image blur. This system will work for most brands or model of drone, regardless of the flight mission software used. It allows the user to identify missing or blurry images in a short time, providing the opportunity for users to re-fly a mission while still on site, ensuring total coverage of an area of interest and the successful future production of an orthomosaic. This quality testing can prevent failed data collection, save the user much time and money in avoiding the need to re-visit field sites, and ensure delivery on ecological research contracts.

## 2. Materials and Methods

We collected drone imagery to build high-resolution maps to track changes in vegetation and flooding in the Chobe Region, northern Botswana. Images were collected with the use of Pix4D capture [19], flying at a height of ~100 m, speed of ~5 m/s and along transects with a front and side overlap of 75% with a DJI Phantom Advanced quadcopter. Images were transferred from the drone secure digital (SD) card and onto a laptop.

The complete image set was used to represent a successful drone flight for the purpose of creating an orthomosaic, labelled "complete". We duplicated this set of imagery and re-

moved 25% of images by deleting files, and used this image set to represent an unsuccessful drone flight, assigning them to a separate folder hereto referred to as "incomplete".

In the software package R [25], we used the exifr package [26] to extract the exchangeable image file format (exif) data of each image allowing for the geographic coordinates hard coded into each image to be extracted. We then plotted the image coordinates, allowing the user to ensure images were collected at even spaces across the area. Further, image coordinates were exported as a .csv file, allowing for the plotting of image location coordinates in a geographic information system over satellite imagery or other layers. This is an additional step that allows the user to easily visualise coverage of the area of interest.

As a further check of dataset quality, we called on the package magick [27] to rank the images (or a subset of images) based on intensity change among pixels of the greyscale image. Images with less edges, such as blurry images or smooth surfaces, have low image variance, while sharp images with many detailed edges return higher image variance [27], a useful technique for many advanced image processing purposes. Rather than define an arbitrary cut-off, the code outputs the 10 images with the lowest variance for the pilot to review and assess their suitability for generating a mosaic. As the example dataset had minimal blurring, we used the magick package [27] to introduce blur in 7% of the images, mimicking potential blur in drone datasets. We then re-ran the algorithm to test the code's ability to detect blurred images in a dataset.

We provide the code on GitHub https://github.com/RoxFrancis/Offline_drone_imagery_assessment (accessed on 3 November 2022) and encourage users to make this quality check a standard procedure for the collection of drone imagery in remote locations.

Orthomosaics presented for visual purposes in the results were produced using Pix4DMapper software [28].

## 3. Results

When plotted, the full set of imagery collected (432 images) showed largely consistent balanced spacing between images captured, and covered the whole area of interest (Figure 1).

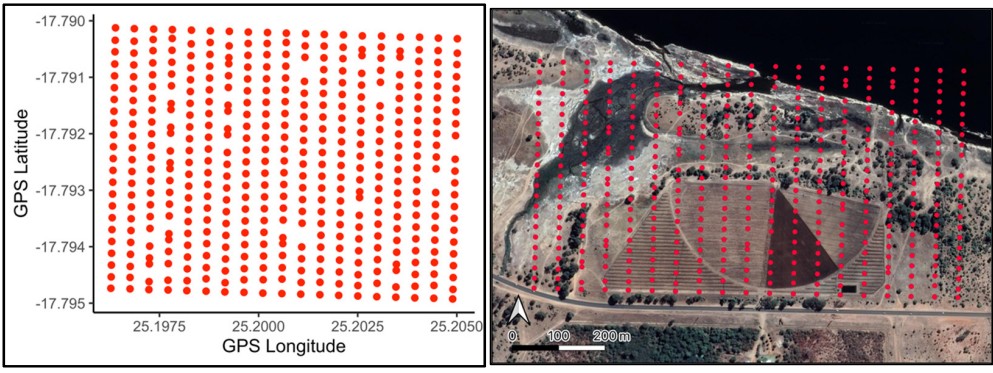

**Figure 1.** GPS locations of drone imagery collected adjacent to the Chobe River, Botswana were extracted and plotted in R for easy inspection of approximate spacing between images, before being exported as a .csv file and plotted over satellite imagery, ensuring total coverage of the area of interest in the complete dataset.

There were a few points where the image was missed or captured late, shown by the grouped points, but the image spacing is largely uniform. As a result, a high-resolution orthomosaic could be produced covering the intended area (noting large expanses of water will normally drop from the orthomosaic process due to few tie points) (Figure 2).

In contrast, the mosaic created from the "incomplete" imagery set (322 images) did not cover the entire area of interest, with inconsistent spacing between images (Figure 3).

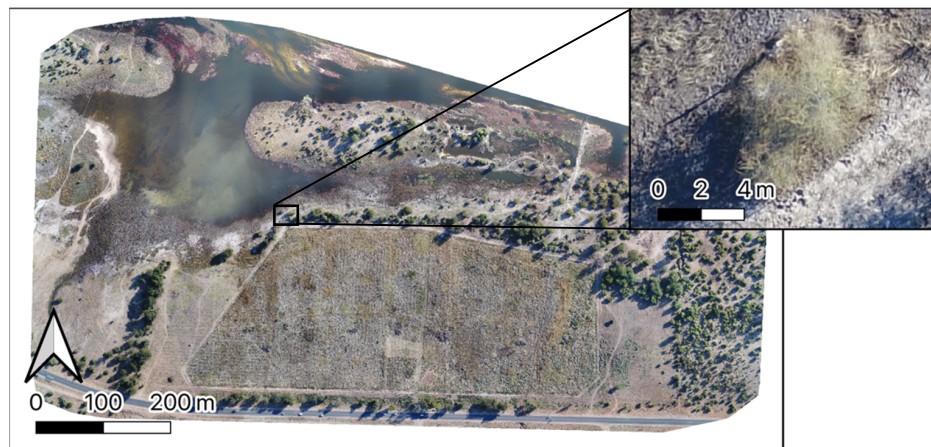

**Figure 2.** The orthomosaic developed from the complete set of images collected alongside the Chobe River, Botswana, showing the high clarity of the orthomosaic when zoomed in (inset).

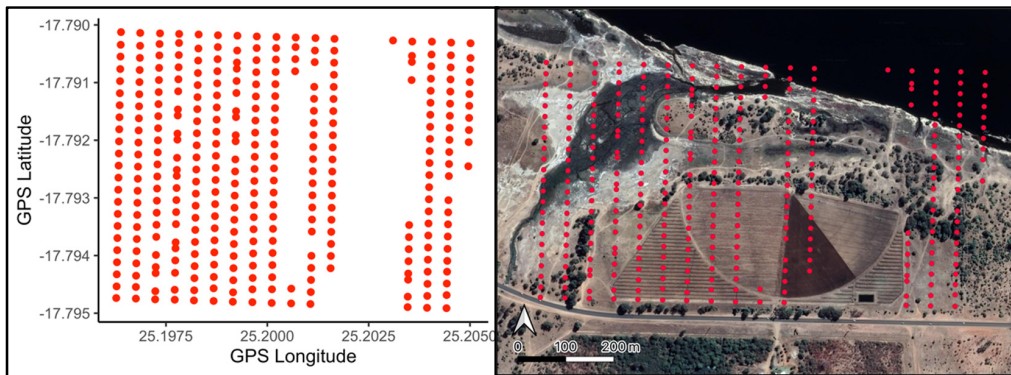

**Figure 3.** GPS locations of a subset of drone imagery collected adjacent to the Chobe River, Botswana, were extracted and plotted in R for easy inspection of approximate spacing between images (left), before being exported as a .csv file and plotted over satellite imagery (right), highlighting gaps in the coverage of the area of interest in the "incomplete" dataset.

This missing data could occur due to software or hardware malfunction, or human error. As a result of the missing images, it did not produce an acceptable mosaic (Figure 4) and would not be useful for further analysis.

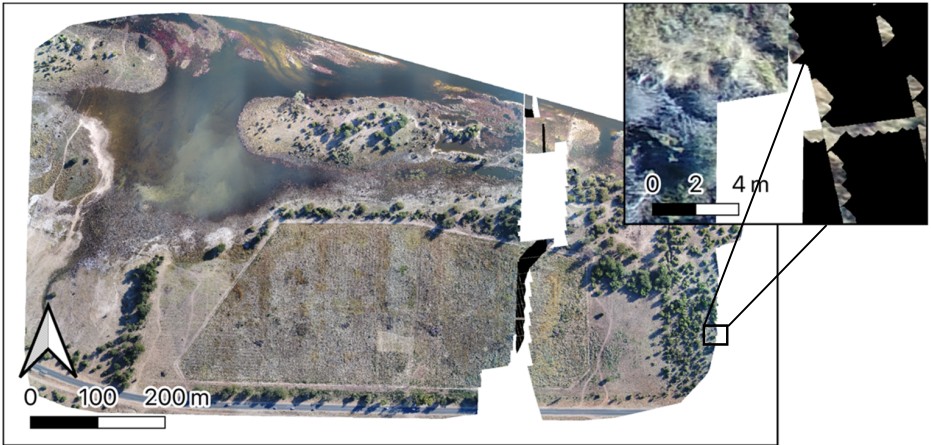

**Figure 4.** The orthomosaic developed from the incomplete set of images collected alongside the Chobe River, Botswana, showing the distortion and gaps in the orthomosaic.

The image dataset had minimal blurring, so instead of blurred images, the output of the variance analysis consisted of large smooth surfaces, such as water (Figure 5). All artificially blurred images which we introduced to test the effectiveness of the algorithm were detected in the top 10 blurred images (Figure 5).

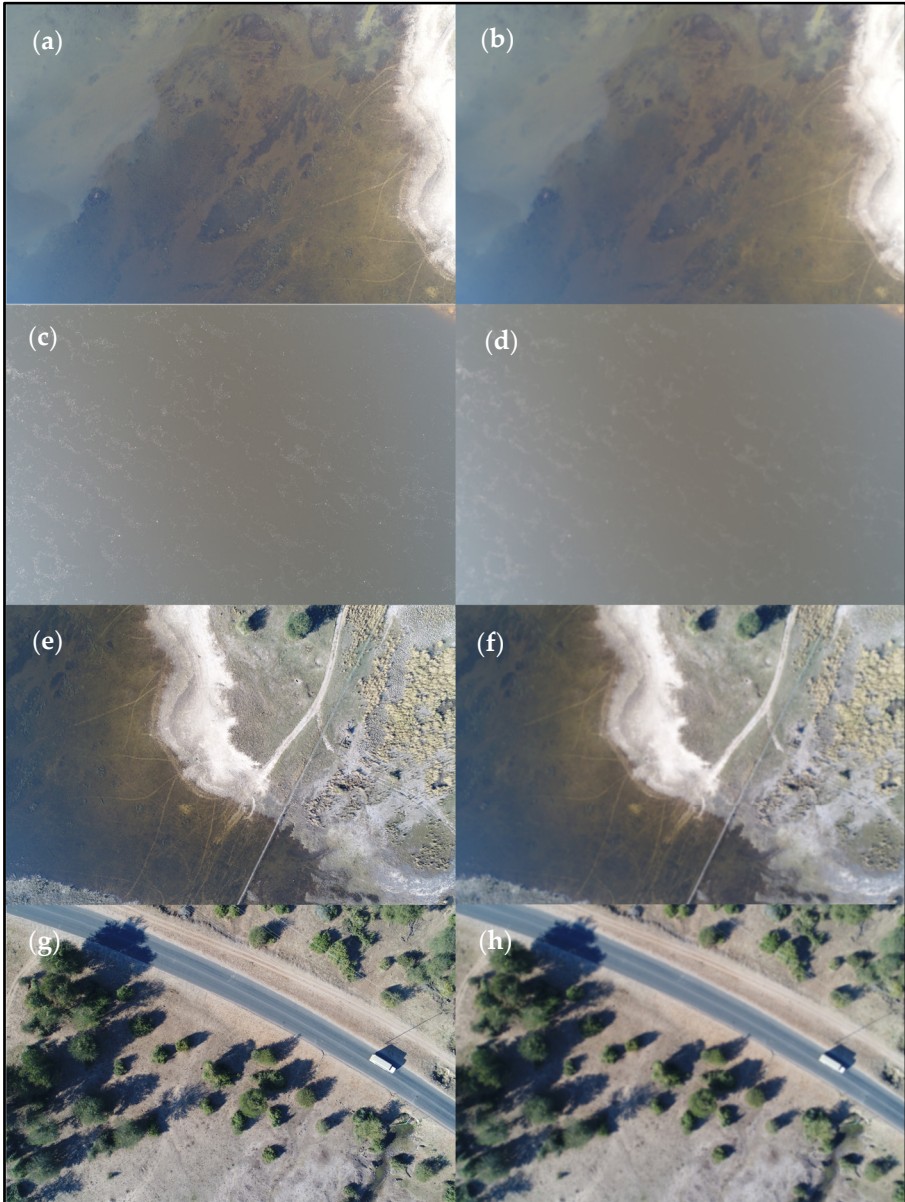

**Figure 5.** Images ranked with the lowest variance from a sample of 10% of the incomplete dataset. Image (**a**) represents the image with the lowest variance of the sample (variance = 14) showing open water, next to its artificially blurred counterpart (**b**), (**c**) represents the second ranked (variance = 26.5) showing texture on the water surface, next to its artificially blurred counterpart (**d**), (**e**) is the ninth ranked (variance = 154) showing water and land, next to its artificially blurred counterpart (**f**) and finally (**g**) is the 10th ranked (variance = 161) showing the smooth ground surface but more texture in the trees, next to its artificially blurred counterpart (**h**).

## 4. Discussion and Conclusions

Flying drones in remote locations poses a unique set of challenges for successful image collection, particularly locations without internet connectivity. For many remote locations, the uploading of imagery to a cloud or server for orthomosaic production is not possible. Pilots therefore risk returning from the field only to find their imagery

was not successfully or entirely collected. Our methods provide a completely offline, open source and free solution for the checking of drone imagery while in the field, that works for most drone brands, models and flight mission software, preventing missed data collection opportunities.

The collection of suitable imagery for the creation of a mosaic is dependant on many factors, such as height, speed and most importantly, transect overlap. This work does not intend to describe ideal drone survey techniques which have been described elsewhere [29–31], but rather to find any mishaps in the collection of the imagery that may have occurred outside of the users control, and provide the opportunity for repeated data collection when necessary. Our method provides a peace of mind check up on the imagery, and can guide a user as to whether they will have a complete and useable dataset.

It is important to note that a complete dataset, that plots with even spaces between images, and covers the entire area of interest, will still not produce a successful orthomosaic if the overlap between transects is not sufficient. Importantly, "sufficient" overlap is project dependant, and will depend greatly on camera and flight statistics, research goals and image background [32]. Further, a complete dataset with sufficient overlap can be useless if each image collected is highly blurred, too light or dark, or the camera gimbal is not correctly calibrated and orientated [17]. Visual inspection of individual images will quickly identify such problems [33], and our code can help to prioritise images that might need attention due to blurring. Investigating images by rank of lowest variance is more time efficient than randomly checking images, which may be unlikely to chance upon potential blur.

Similar industry-specific advances in the processing of drone imagery are arising, such as the repeated temporal alignment of agricultural plots to explore plant development [34] and the development of end to end software packages with multi-sensor functions [35]. Such works highlight the constant improvements in not only drone technology but pre-and post-processing software, which will continue to facilitate the use of drones in more and more industries.

Within this work, we provide two example datasets to highlight to a user the visual differences between a successful and unsuccessful flight mission, and resultant orthomosaics (https://figshare.com/articles/media/Complete_and_Incomplete_Image_Collections/20479215, accessed on 3 November 2022). The "complete" dataset produced the best orthomosaic (Figure 3), and any errors in the orthomosaic produced from the "incomplete" dataset (Figure 4) are due to the missing images, as all other processing options remained the same. The image blur quantification and ranking returned crisp images of smooth surfaces (Figure 5), so is not a concern for production of an orthomosaic from this dataset. The code was effective at finding the artificially blurred images, proving its usefulness on other users' datasets.

While our code does require the movement of images from the drone to a laptop computer—the benefit of this work is that it is applicable to most brands and models of drones, controllers, flight mission software, photogrammetry software, storage medium or computer. It requires only a few minutes to extract the exif data (depending on the number of images) using exifr [26], making it suitable for checking in the field. The few cases where this piece of code will not work however, is when exif data are not saved within the images collected, but are rather saved in an external image geolocation text file.

We have found that the time checking data quality immediately after the mission is time well spent. A particularly useful application of this code is when an area of interest is to be estimated in the field. Many ecological applications require the in-field determination of a boundary, such as covering the extent of bird colonies which have regularly changing boundaries [11,36].

Future improvements on this work could include automated detection for image brightness boundaries. Such work would be largely project specific and dependant on the area and object of interest. For example, when working with a dark coloured species on a dark background such as water it might be beneficial to have lower limits on image brightness. The inverse of this scenario could be counting white birds on sand or ice [37],

where over-exposure is more likely to be an issue, and upper limits on brightness could be specified. Further, automated detection of missing images is a potential addition to the code. Such work would require distance calculations between neighbouring image locations, but these could be elevated by edge photographs which have no neighbouring image. Such edge images could be removed from the calculations when working with square or rectangular flight missions, but irregular areas (such as those often covered in ecological surveys) would complicate the identification of "edge" images.

The successful creation of orthomosaics is important for the monitoring and analysis of ecological trends [38]. This simple in-field quality assurance check of collected drone imagery can help to ensure successful mosaic creation, contributing to the success of research projects, and delivery on ecological contracts.

**Author Contributions:** Conceptualization, R.J.F., J.A.M.; methodology, R.J.F., J.A.M.; software, R.J.F., J.A.M.; validation, K.J.B.; resources, K.J.B.; writing—original draft preparation, R.J.F., J.A.M.; writing—review and editing, R.J.F., J.A.M., K.J.B.; visualization, R.J.F.; supervision, K.J.B.; project administration, R.J.F.; funding acquisition, K.J.B. All authors have read and agreed to the published version of the manuscript.

**Funding:** This research received financial support from Taronga Conservation Society, the Australian Commonwealth Government Research Training Program and the Centre for Ecosystem Science, University of New South Wales Sydney.

**Data Availability Statement:** Code to run imagery checks can be found on GitHub https://github.com/RoxFrancis/Offline_drone_imagery_assessment (accessed on 3 November 2022) and we provide a practice imagery data set on Figshare https://figshare.com/articles/media/Complete_and_Incomplete_Image_Collections/20479215 (accessed on 3 November 2022).

**Acknowledgments:** This research received financial support from Taronga Conservation Society, the Australian Commonwealth Government Research Training Program and the Centre for Ecosystem Science, University of New South Wales Sydney. This study was conducted under the guidelines of the UNSW Animal Care and Ethics, permit 13/3B. We also thank the Government of Botswana for access to research permits EWT 8/36/4 XXIV (179), and drone permit RPA (H) 211.

**Conflicts of Interest:** The authors declare no conflict of interest.

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
