# Peer review of "Offline Imagery Checks for Remote Drone Usage"

_drones, doi:10.3390/drones6120395_

Round 1
Reviewer 1 Report
The authors present a technical Communication for a solution to a common problem in the application of drones and modern image processing pipelines for field remote sensing - lack of connectivity to cloud-based software and therefore lack of confidence in field collected data products. The Communication is well written, with appropriate formatting, headings, background, and section content. Figures are appropriate and need light copy-editing, but that is not the author's responsibility. In all, I think this Communication is an appropriate contribution to the journal Drones, I personally have conducted many missions in heavy winds and would have gotten a lot of value out of such a tool to identify gross errors in flight tracks before leaving the site! However the authors should make light corrections to reframe the purpose of their work and the breadth of the current problem.
- Abstract and Introduction
- the authors make the case that drone software is "created for urban use" which implies real-time or close proximity to internet connectivity, and which therefore presents a problem for non-urban settings where connectivity is low.
- However this is "marketing", not origin. The authors should reframe their argument to focus on the closed-source, propriety, and cloud-based nature of these software platforms, which does speak to the nature of their creation and not marketing to urban vs. rural areas. Indeed, many drones are marketed for rural and agricultural use in the developed world, with little to no regard for connectivity. It is also not hard to find marketing articles by DroneDeploy and other vendors about work in the rural or developing world (e.g., "How OpenStreetMap Uganda Uses DroneDeploy").
- Similarly, the authors do not address open source community photogrammetry solutions (For example OpenDroneMap - WebOpenDroneMap) which can be local or cloud-based.
- Even so, the authors' case is still valid: cloud-based solutions alone make it difficult to have confidence that a data collection met sufficient quality standards before leaving the field. Confidence in data quality before leaving the field is tremendously important considering the costs of field work and mobilizing, whether in rural areas of the developed world or other remote locations.
47-49 reframe this to be "for the purposes of evaluating flight quality" or something to that effect.
61-67 This is not just a problem for cloud-based solutions. Processing can also take several hours or even days making it difficult to understand collection quality. please mention this.
82 you are not plotting "drone imagery" but the GPS location of the drone when an image was taken. rephrase this, for example "drone image geotags", "image location" or something to that effect.
83 "work with any brand of drone", assuming that the drone geotags its photos in the image EXIF metadata. Some actually do not, and generate an image geolocation text file to use in post-processing.
96 "random": the patterns shown in Figure 3 do not suggest "random" removal, but a more systematic approach to illustrate the tool's capability. please rephrase the method accordingly.
106-112 what is the source of this method for detecting blur? is this original work or a common computer vision technique, please reference.
110 reference not in journal standard format.
113-114 please provide the github link here. Also, drones allows for supplements and appendices. since the source code is a small R file, please include as a supplement with the manuscript
177-178 please introduce blur manually into some images and demonstrate how the blur filter would detect and return these. This would be an appropriate simulation of results from the single, high quality dataset. for example, if you blurred 1% of the images, 4-5 frames, would the algorithm register those? please produce a graphic of the 10 low variance image results with and without blurring introduced to demonstrate the effectiveness of the blurring algorithm. As is, the readers have no conclusive test of this part of the tool.
192 please include the free and open source license with the code, in the text header of the source code, for example, GPL, Creative Commons, etc. at the authors' discretion.
204 In the discussion, could you discuss how your code might be advanced to detect images too bright or too dark, for a future release?
214-219 please link the datasets here, the link at the acknowledgements is broken.
220-222 only if the drone assigns exif to the imagery, some devices actually do not do this. please address by mentioning this requirement of the code.
220-227 Error detection from the code is still manual - a visual inspection of the output geotag map. Are their opportunities for a future to release to detect or highlight areas of potential bad overlap? for example the missing flight lines in Figure 3 or the skipped / misfire images that caused small gaps. Perhaps suggest this.
Reviewer 2 Report
This manuscript developed the code to plot drone-captured photos and assess the necessary image collection requirements for the reliable production of a mosaic. There are several points that need further exploration.
1. There are lots of long sentences that make the manuscript hard to read. Please split them into short sentences.
2. There are some typo errors, for example, in line 129 the "GPS Locations" should be "GPS locations".
2. The full name of QAQC in the abstract should be given.
3. Please summarise the main contributions in the introduction parts.
4. There are a lot of sentences that mention "sufficient quality". How to define sufficient quality?
5. More description of the method used in this work should be mentioned rather than only some references.
6. The drone takes continuous images. How to choose the images with the proper overlap rate should be given.
7. How to process the overlap areas to stitch images should be provided.
8. How to define successful mosaicking of imagery and comparison with other related works should be given to demonstrate the effectiveness of the proposed method
Reviewer 3 Report
The idea of ​​assembling orthomosaics without internet connection is interesting. However, most software systems that come with commercial drones already have applications for assembling orthomosaics. Often, in order to carry out autonomous flights, one already has extensive knowledge of the region to be photographed, including satellite photos of these regions.
Round 2
Reviewer 1 Report
Content revisions are acceptable, thank you.
Because of track changes, I cannot speak to the document format and body, but i believe that will be addressed.